# Phytochemicals and Vitamin D for a Healthy Life and Prevention of Diseases

**DOI:** 10.3390/ijms241512167

**Published:** 2023-07-29

**Authors:** Kazuki Santa, Kenji Watanabe, Yoshio Kumazawa, Isao Nagaoka

**Affiliations:** 1Department of Biotechnology, Tokyo College of Biotechnology, Ota-ku, Tokyo 114-0032, Japan; kazuki_santa@hotmail.com; 2Center for Kampo Medicine, Keio University, Tokyo 160-8582, Japan; 3Yokohama University of Pharmacy, Yokohama 245-0066, Japan; 4Vino Science Japan Inc., Kawasaki 210-0855, Japan; 5Department of Biochemistry and Systems Biomedicine, Juntendo University Graduate School of Medicine, Bunkyo-ku, Tokyo 113-8421, Japan; 6Faculty of Medical Science, Juntendo University, Urayasu 279-0013, Japan

**Keywords:** ME-BYO (mibyou), lifestyle-related diseases, lung disorders, sepsis, microbiota, phytochemicals, vitamin D

## Abstract

A variety of phytocompounds contained in medical plants have been used as medication, including Kampo (traditional Japanese) medicine. Phytochemicals are one category of the chemical compounds mainly known as antioxidants, and recently, their anti-inflammatory effects in preventing chronic inflammation have received much attention. Here, we present a narrative review of the health-promotion and disease-prevention effects of phytochemicals, including polyphenols, the latter of which are abundant in onions, oranges, tea, soybeans, turmeric, cacao, and grapes, along with the synergetic effects of vitamin D. A phenomenon currently gaining popularity in Japan is finding non-disease conditions, so-called ME-BYO (mibyou) and treating them before they develop into illnesses. In addition to lifestyle-related diseases such as metabolic syndrome and obesity, dementia and frailty, commonly found in the elderly, are included as underlying conditions. These conditions are typically induced by chronic inflammation and might result in multiple organ failure or cancer if left untreated. Maintaining gut microbiota is important for suppressing (recently increasing) intestinal disorders and for upregulating immunity. During the COVID-19 pandemic, the interest in phytochemicals and vitamin D for disease prevention increased, as viral and bacterial infection to the lung causes fatal inflammation, and chronic inflammation induces pulmonary fibrosis. Furthermore, sepsis is a disorder inducing severe organ failure by the infection of microbes, with a high mortality ratio in non-coronary ICUs. However, antimicrobial peptides (AMPs) working using natural immunity suppress sepsis at the early stage. The intake of phytochemicals and vitamin D enhances anti-inflammatory effects, upregulates immunity, and reduces the risk of chronic disorders by means of keeping healthy gut microbiota. Evidence acquired during the COVID-19 pandemic revealed that daily improvement and prevention of underlying conditions, in terms of lifestyle-related diseases, is very important because they increase the risk of infectious diseases. This narrative review discusses the importance of the intake of phytochemicals and vitamin D for a healthy lifestyle and the prevention of ME-BYO, non-disease conditions.

## 1. Introduction

A variety of phytocompounds have been used as medicines in traditional medicine. Since ancient times, many drugs have been created from plant-derived ingredients, similarly to how antibiotics were developed from the research of bacteria. Aspirin is a typical plant-derived drug, isolated from willow trees [1], and currently, other plant-derived medicines such as artemisinin, an antimalarial drug derived from wormwood leaves, have been widely used [2]. *Kampo* (traditional Japanese) medicine is a good example of the use of such plant-derived ingredients; it was introduced from ancient mainland China before the spread of Western medicine in Japan in the late 18th century [3], and recently, a variety of plant-derived ingredients have been used for the promotion of a healthy lifestyle [4]. Furthermore, the COVID-19 pandemic has rekindled interest in the activities of natural substances for managing clinical outcomes. There are many kinds of plant-derived ingredients; however, recently, attention has focused on a group of plant-derived chemicals termed phytochemicals. Phytochemicals generally consist of various classes of compounds, including terpenoids, carotenoids, flavonoids (polyphenols), etc. [5]. Here, we describe the effects of phytochemicals, including the polyphenols abundantly contained in onions, citrus fruits, tea, soybeans, turmeric, cacao, and grapes—familiar vegetables and fruits in Japan—combined with vitamin D, an attractive synergic formula for a healthy lifestyle and the prevention of diseases.

Recently, the concept of ME-BYO (mibyou) has entered the public focus. This idea comes from ancient textbook, “Huangdi Neijing”, which states that “saints cure pre-illness instead of manifest illness“, emphasising the importance of the primary prevention of disease, as based on 4000-year-old Chinese medical science. This concept is now widely recognised in Japan, accompanied by a rise in health-consciousness [6]. We cannot distinguish cleanly between the healthy state and disease. ME-BYO (non-disease state) is the state continuously changing from healthy state to disease. Underlying conditions that received great attention during the COVID-19 pandemic, as well as lifestyle-related diseases such as metabolic syndrome, obesity, dementia, and frailty, often trigger serious consequences. Therefore, we need to be cautious about these conditions, and their amelioration is necessary. Parts of these conditions can be regarded as ME-BYO. Chronic inflammation is the cause of the most of these disorders, and gut dysbiosis is related to intestinal disorders, which are increasing in younger generations. Furthermore, these conditions are strongly correlated with what we eat [7]. The combined administration of phytochemicals and vitamin D modulates gut microbiota, activates regulatory T cells (Treg), upregulates intestinal immune responses, and protects us from various disorders.

According to the WHO, in 2022, the second-most-deadly infectious disease after COVID-19 was tuberculosis. Pneumonia is still the greatest cause of death in elderly people. More than 95% of deaths from tuberculosis are in low- and middle-income countries where investment in disease prevention and public health are low. The United Nations has requested that each country exert greater efforts to end the tuberculosis epidemic by 2030, as one of the U.N. Sustainable Development Goals (SDGs) [8]. Acute lung injury (ALI) such as COVID-19 induces extensive lung inflammation and finally causes fatal respiratory failure [9]. In addition, chronic-inflammation-induced lung disorders such as interstitial pneumonia gradually damage lung alveoli, and COPD sometimes manifests as a sequela of COVID-19; both of these disorders are becoming major problems [10,11]. Furthermore, sepsis is multiple organ failure induced by the infection of microbes such as bacteria and viruses. Systemically induced or organ-based cytokine storms cause disseminated intravascular coagulation (DIC) [12]. Recently, it was revealed that LL-37, an antimicrobial peptide (AMP) modulating the natural immune response, suppresses sepsis via the neutralisation of LPS activities and microbial infections [13]. The administration of vitamin D also increases the amount of LL-37, and is associated with the modulation of the immune system.

A combination of phytochemicals and vitamin D has been reported to reduce the risk of these disorders by ameliorating chronic inflammation through the suppression of TNF-α production [14]. This narrative review discusses the potential of phytochemicals alongside vitamin D for disease risk reduction, promoting a healthy lifestyle, and the prevention of ME-BYO.

## 2. Brief Methodology of the Review Article

Briefly, we explain the methodology of the search method of scientific papers in the references of this review. This review is narrative review, as we described above. We searched for papers in PubMed with combined keywords “phytochemicals”, “vitamin D” and/or the name of each phytochemical, such as “polyphenol” and “quercetin”, as well as the names of disorders, such as “lifestyle-related diseases”, “gut microbiota”, “lung disease” and/or “sepsis”. After the search, we composed this review article in reference to the papers we considered necessary. Furthermore, over 50% of references have been published within 5 years; others were thought important or were well-known papers in the field. The references include our previous publications, and we have searched and collected scientific papers that are especially attractive in our region, including papers introduced in the press releases of universities in Japan, webpages from companies and institutions, and papers discussed in web-news-featured special articles in related fields from newspapers. The references in this review include many “review” papers, as they summarise many research findings.

In Section 3, we summarise the experiment-based information from experiments conducted in the referenced papers regarding phytochemicals contained in fruits and vegetables in Table in Section 3. We searched on PubMed for the keywords “grape and polyphenol” with article types “Clinical trial and Randomised Controlled Trial” across two years, from 2021 to 2022, in addition to 2023 before this research effort. As a result, 16 search results emerged, and they are summarised in Table in Section 3.7. Then, Table in Section 4.1 shows references organised by diseases in Section 4 and the benefits of phytochemicals and vitamin D consumption. Since the Table in Section 4.1 contains some references shown in Table in Section 3, the item “effects” was deleted, and they were re-sorted by “diseases”. Since there are many review articles in the reference papers, references that did not include detailed experimental methods or doses were excluded from the table.

## 3. Phytochemicals and Vitamin D

The term phytochemicals originates from the Greek, signifying “plant chemicals”. Phytochemicals are produced by plants to protect themselves from UV and other predatory presences. These chemicals range from toxins to drugs used for traditional medical purposes. Here, we explain phytochemicals well known in Japan, mainly those with anti-inflammatory properties contained in fruits and vegetables that are called phytonutrients, and attractive to use for the prevention of diseases and promotion of a healthy condition [15]. Nowadays, in addition to the three major nutrients (sugars, lipids, and proteins), Japanese nutritionists tend to regard minerals and vitamins the fourth and fifth major nutrients, with dietary fibres being the sixth. In addition, some nutritionists tend to regard phytochemicals as the seventh nutrient, and the recognition of their importance has been increasing [16].

Among phytochemicals, polyphenols are a generic name of phytocompounds containing phenolic hydroxyl groups within their chemical formulas. They comprise over 5000 pigments, and are bitter-tasting compounds produced by photosynthesis and present in most plants [17]. Typical polyphenols are catechins, anthocyanins, rutin, and isoflavones, which are classified as flavonoids, and others are curcumin and the polyphenols contained in cacao beans. Flavonoids are a generic term for plant secondary metabolites containing a flavan basic skeleton [18]. Generally, terpenoids, carotenoids, polyphenols (including flavonoids), and sulphuric compounds tend to be classified as phytochemicals.

Below are the vegetables and fruits containing phytochemicals considered attractive for their health-promoting effects. Table 1 summarises the potential effects of phytochemicals and vitamin D. In addition, we discuss the combined effects of phytochemicals and vitamin D.

### 3.1. Onions

Quercetin is a representative flavonoid phytochemical found in onions (*Allium cepa*) and mainly contained in onion peel rather than the bulb [19,51]. The use of onion skin powder in food is increasing as of late as a pathway for the intake of phytochemicals for their health benefits. Quercetin is the strongest known anti-inflammatory flavonoid. In addition, quercetin is contained in the ingredients of many herbal medicines that feature in *Kampo*, such as Japanese hawthorn (*Crataegus cuneata*) [52]. Furthermore, onions are rich in rutin, which is a glycoside form of quercetin [20]. Allicin is another well-known phytochemical, contained in onions, garlic (*Allium sativum*), and leeks (*Allium fistulosum*), which is one of the allyl sulphides that characterises the spiciness and smell of onion. Figure 1 shows the chemical structure of quercetin. Onions have been used as a medical plant since ancient times, and their health benefits are still attractive [53].

### 3.2. Citrus Fruits

Citrus fruits and oranges contain large amounts of phytochemicals. These are mainly contained in the skin and especially in the mesocarp instead of the pulp. Japanese mandarin (Unsyu-mikan) (*Citrus unshiu*) is the most popular citrus fruit in Japan, and its citrus peel, termed “chinpi”, is used to a high degree in Kampo medicine. To make chinpi, orange peel needs to be dried for over 1 year; its relevant phytochemicals include hesperidin, naringin, rutin, etc. [54]. Below are the well-known orange phytochemicals. They are not only used for pharmacological purposes in the manner of traditional medicines, but also are ingested in food through the use of healthy ingredients.

Hesperidin is mainly contained in Japanese mandarins and Hassaku oranges (*Citrus hassaku*) and reduces cholesterol and blood pressure, prevents bone density loss, and shows protection against sepsis. Hesperidin is poorly soluble in water and, owing to its low absorption ratio in the body, hesperidin glycoside, a glucose-conjugated form, is frequently used [21,22]. β-cryptoxanthin is a xanthophyll carotenoid contained in a variety of citrus fruits such as the Japanese mandarin. β-cryptoxanthin is a provitamin A converted into vitamin A and is known to possess an antioxidant effect effective against liver disorders, arteriosclerosis, diabetes, and osteoporosis; in promoting bone metabolism; and in the prevention of cancer [55]. Naringin is a bitter component contained in oranges, grapefruits (*Citrus paradisi*), daidai (*Citrus aurantium*), Natsu-mikan (*Citrus natsudaidai*), etc., and is contained in some herbal medicines made from them. Naringin is effective in its anti-oxidative properties, obesity prevention, treatment of hypertension, and for lowering fasting blood sugar levels [23]. Nobiletin is contained in ponkan (*Citrus poonensis*) and shiikwaasa (*Citrus depressa*). It has anti-inflammatory effects, promotes skin metabolism, suppresses blood sugar levels by way of adiponectin production, reduces the size of enlarged adipocytes, burns fat, prevents chronic inflammation by suppressing histamine release, and prevents dementia resulting from Alzheimer’s disease [24,25]. Rutin is quercetin glycoside, a flavonoid glycoside found in *Ruta graveolens*. It is also found in some citrus fruits, including Japanese mandarins, grapefruits, lemons (*Citrus limon*), and limes (*Citrus aurantiifolia*), as well as buckwheat (*Fagopyrum esculentum*), and is used as a vascular protection medication in many countries [26]. Recently, Sudachitin, which is present in Sudachi (*Citrus Sudachi*), has been shown to regulate circadian rhythm and suppress the accumulation of fat in the liver [27].

### 3.3. Tea

Tea (*Camellia sinensis*) contains catechins; raw tea leaves contain epicatechin (EC), epigallocatechin (EGC), epicatechin gallate (ECG), and epigallocatechin gallate (EGCG). When heated, these change to catechin (C) and gallocatechin (GC), catechin gallate (CG), and gallocatechin gallate (GCG). Catechins have various physiological activities, controlling blood pressure, cholesterol, and sugar levels, and possessing anti-oxidative and anti-ageing effects [56]. An enhanced anti-obesity effect from the combined administration of citrus polyphenols and tea catechins has been reported [28]. The effects of catechins for viral inactivation are well known, and recent research has shown the effect of EGCG in the inactivation of SARS-CoV-2 virus [29,30]. Catechins from tea leaves have lipolytic properties and reduce the accumulation of visceral fat [31,32]. In addition, theaflavin, a catechin polymer, is produced during the fermentation process from green tea to black tea and has anti-inflammation, anti-oxidation, anti-cancer, and anti-obesity properties [33]. In addition, epidemiological studies have shown that catechins prevent dementia [34]. As we described above, catechins contained in tea leaves have various healthy properties.

### 3.4. Soybeans

Soybeans (*Glycine max*) contain isoflavones such as genistein and daidzein that work as oestrogen-like substances, that is, as oestrogen receptor agonists [35]. The oestrogenic effect of soy isoflavones comes from equol. The conversion of equol from isoflavones requires the metabolism of gut microbes, but the population of people who have gut microbes that can convert isoflavone to equol is limited by region and age [36,37].

### 3.5. Turmeric

Turmeric (*Curcuma longa*) is known as a spice used in Indian cuisine and is quite familiar to Japanese people, as it is the essential ingredient of Japanese curry rice. The turmeric rhizome contains a large amount of the polyphenol curcumin. Spices such as curcumin are known to upregulate immunity, and its health-promoting abilities have proven attractive around the world [57]. Turmeric has anti-inflammatory, anti-oxidative, and anti-cancer properties, and much research has shown that curcumin has protective effects for a variety of disorders [58]. In addition, curcumin is well known for its effect in enhancing liver function [38].

### 3.6. Cacao

Cacao (*Theobroma cacao*) is a raw material for chocolates and cocoa drinks consumed all over the world. Cacao phytochemicals mainly consist of polyphenols that have been confirmed as anti-inflammatory and anti-oxidative and ameliorate the effects of metabolic-syndrome-related disorders, similarly to other polyphenols contained in plants [59]. Phytochemicals in cacao beans are called cacao polyphenols and consist of the family of flavan-3-ols (flavonoids), comprising epicatechin, catechin, and their polymers, the procyanidins [60,61]. A recent study investigated the mechanism of cocoa flavanols in ameliorating metabolic-syndrome-related disorders [39]. Another study has shown the preventive effects of cacao polyphenol against blood sugar spikes via the enhancement of GLP-1 and insulin production [40].

### 3.7. Grapes

Grapes (*Vitis* spp.) are one of the food crops with the longest histories. They were cultivated for wine production rather than for food in ancient times. The polyphenol resveratrol is a well-known grape phytochemical, which made headlines in the form of the French paradox. In this phenomenon, phytochemicals contained in red wine have been reportedly associated with there being fewer heart disease patients in France compared with America, making polyphenols widely known to the public [41]. Grape phytochemicals are mainly contained in the skins or seeds and are classified into terpenoids, carotenoids, and flavonoids. The terpenoids include oleanolic acid, which exists on the surface of the fruit; the carotenoids include β-carotene; and the flavonoids are subdivided into flavon-3-ols including quercetin, flavan-3-ols including catechins, and anthocyanins including malvidin [62]. Tannins and procyanidin, which are polymers of catechins, are rich in grapes as well.

Quercetin in grapes is contained in the skin and seeds, but mostly in the skin. Quercetin glucoside, a water-soluble glycoside, is decomposed into quercetin by bacteria in the large intestine and is absorbed in the gut. After absorption, quercetin conjugates again with glucuronic acid to become quercetin glucuronide and circulates into the bloodstream. Quercetin has anti-inflammatory properties, but it requires an activation process. When blood vessels induce inflammation, macrophages gathered there secrete β-glucuronidase. This enzyme cuts glucuronic acid from quercetin glucuronide to make the active form of quercetin. This activated quercetin suppresses the production of TNF-α and ameliorates inflammation [42].

Anthocyanins are water-soluble pigments, of red, blue, and purple colour, widely found in the flowers and fruits of plants [63]. Anthocyanins, abundant in grape skin and also found in grape seeds, mainly consist of malvidin, cyanidin, and delphinidin. They have anti-inflammatory and anti-oxidative effects and reduce the incidence of cardiovascular diseases and type-2 diabetes, as well as the mortality rate [43]. Procyanidins are oligomers and polymers of epicatechins or catechins which are abundantly present in cacao beans as well. Recently, procyanidins have gained much attention for their anti-ageing activities attained through the suppression of senescent cell accumulation, which causes the senescence-associated secretory phenotype (SASP) [44]. In addition, procyanidins are dose-dependently associated with the gut–brain axis in the central nervous system via neurotransmitter receptors [45]. Furthermore, oleanolic acid is a component of the white powder, called bloom, found on the surface of grapes. Japanese hawthorn contains a large amount of oleanolic acid as well. This compound activates intestinal peristalsis as an agonist of the bile acid receptor TGR5 [46]. In addition, oleanolic acid suppresses cholestasis and releases liver cholesterol into the intestine as bile acids, ultimately excreting them from the body [64]. Grape phytochemicals have already been used for medical purposes and in functional foods such as grape seed extract (GSE) [47,48].

Table 2 shows the clinical trials related to grape polyphenols. In addition to cardiovascular disorders, a recently conducted variety of randomised controlled trials for metabolic syndrome related disorders including dementia, gut microbiota related research, periodontal diseases, allergic diseases, and cancer have revealed intriguing findings [65,66,67,68,69,70,71,72,73,74,75,76,77,78,79]. These clinical trials have confirmed the benefits of the consumption of polyphenol phytochemicals for their risk-reducing effects in a variety of disorders.

### 3.8. Health Promotion Effect of Vitamin D and Interactions with Phytochemicals

Vitamins and minerals are classified as two of the five major nutrients in Japanese nutrition science. Vitamins are essential nutrients for the existence of every organism, and most of them are organic compounds that our body cannot synthesise. However, vitamin D is different from every other vitamin as to one point: it is synthesised from cholesterol in skin cells under UV exposure [80]. Vitamin D works as a hormone associated with bone formation and breakdown and its deficiency causes rickets and osteomalacia [81]. In addition, the importance of vitamin D has been further indicated by the fact that its deficiency correlates with a variety of disorders, including hypertension, tuberculosis, cancer, periodontal disease, multiple sclerosis, and autoimmune disorders [82,83]. The intake of vitamin D and phytochemicals was recommended during the COVID-19 crisis to upregulate immunity. Since the combined intake of vitamin D with phytochemicals promotes a healthy lifestyle, vitamin D seems to have an adjuvant effect on phytochemicals [84]. On the other hand, it has been reported that 98% of the Japanese population has an insufficient amount of serum vitamin D (under 30 ng/mL) [85].

The following descriptions are the interactions between vitamin D and phytochemicals: Vitamin D is susceptible to oxygen and high temperature, but quercetin suppresses the decomposition of vitamin D [86]. In addition, both vitamin D and phytochemicals possess anti-inflammatory effects and upregulating properties towards immune responses [87]. They are both important for maintaining healthy gut microbiota [88]. There has been intriguing genetic research showing the benefit of vitamin D and phytochemicals in the remission of inflammatory bowel disease [89]. In a dietary intervention study, they were found to downregulate the level of TNF-α, an important factor in chronic inflammation [90].

## 4. The Idea of ME-BYO (mibyou)

Recently, to emphasise the importance of preventive medicine, the term ME-BYO has become popular in Japan. This word comes from a 2000-year-old ancient Chinese medical book and is the state of not being ill but not being healthy. Figure 2 is a diagram of ME-BYO, indicating the correlation between the onset of disease, ME-BYO (non-disease state), and the healthy state. The importance of understanding the state of ME-BYO has been increasing because continuously preventing the non-disease condition in younger generations eventually leads to reduced medical expenses and the achievement of a healthy and long life. Heart diseases and cerebrovascular diseases are the deadliest diseases in Japan and are involved in lifestyle-related diseases. In addition, patients with two or more risk factors of metabolic syndrome, including visceral fat obesity, diabetes, hypertension, and dyslipidaemia, tend to experience the onset of deadly heart and cerebrovascular diseases. Therefore, these high-mortality diseases can be prevented by removing these risk factors [91,92,93].

According to Japan Mibyou Association, ME-BYO, the non-disease state, is defined as a state in which there are no subjective symptoms, but abnormalities are found during examinations or a state in which there are subjective symptoms, but no abnormalities are found in examinations. Recent research shows that the statistical analysis of biological signals, specifically state-transition-based local network entropy (SNE), can be applicable in the diagnosis of this non-disease state [94,95]. In addition, other research shows that the administration of anti-obesity *Kampo* medicine prevents the onset of disorders and is associated with the improvement of gut microbiota [96]. These findings make it possible to understand the idea of ME-BYO and the importance of disease prevention.

Good sleep, healthy diet, and moderate exercise are important in the prevention of ME-BYO. First, ensuring good sleep during the night is important for many living creatures. In the case of humans, there are two repeated conditions in sleep: rapid eye movement (REM) sleep (light sleep) and non-REM sleep (deep sleep). In REM sleep, the skeletal muscles are relaxed, and our bodies are in rest, but our cerebrums perform the processing of data and memories. On the other hand, non-REM sleep is the state of “brain sleep” where renewal of the body is progressing, including repair of damaged cells and regeneration of tissues, in addition to growth of the body in the growing phase [97]. In addition, the biological clock controls energy metabolism in the cell, but this clock deteriorates with age. To suppress this phenomenon, eating a low-calorie diet, which activates the longevity-related Sirtuin genes, is recommended to keep our cells rejuvenated [98]. Furthermore, avoiding high-calorie diets is important for the prevention of the manifestation of metabolic syndrome, to which end the anti-oxidative and anti-inflammation properties of phytochemicals are useful as well [99]. The Mediterranean diet is a well-known good example of a healthy diet [100,101]. Finally, declining metabolism as the result of a lack of exercise is responsible for the onset of metabolic syndrome and the cause of other conditions such as frailty and locomotive syndrome. Frailty mainly refers to physical decline in the elderly, but it also is demonstrated as a declined capacity for exercise at any age. This condition is caused by sarcopenia (loss of muscles), osteoporosis, and weakness of muscles [102,103,104]. Bones, joints, muscles, and nerves are collectively called the locomotive system, and locomotive syndrome is the condition of the deterioration of the locomotive system [105,106]. The mitochondria that produce intracellular energy in patients with frailty and locomotive syndrome are decreased. Old mitochondria in these patients produce massive amounts of reactive oxides. However, these dysfunctional organelles are depleted by autophagy [107].

Below are the disorders that are expected to be prevented by the intake of phytochemicals and vitamin D whilst a patient is in the ME-BYO state.

### 4.1. Lifestyle-Related Diseases and Underlying Conditions

During the COVID-19 pandemic, older generations had priority for vaccination, followed by patients with underlying conditions, including lifestyle-related diseases. The immune system in the elderly is compromised due to ageing, and the ability to suppress or eliminate viral proliferation is not sufficient. Underlying conditions include chronic heart disease, kidney failure, respiratory problems, immune suppressive disorders such as cancer, and hospital visits or hospitalisation for sleep apnoea syndrome. These procedures were taken because patients with these conditions tend to have weakened resistance against viruses [108]. Severe underlying conditions that require continuous medical intervention may not be included in ME-BYO. However, patients with chronic inflammations such as diabetes tend to produce high amounts of TNF-α, increasing their susceptibility to septic shock, and, if infected with viruses or microbes, their symptoms tend to become serious much more easily [109]. Table 3 summarises the health-promoting effects of phytochemicals and vitamin D against the disorders described below.

Recently, low-metabolism-related disorders such as frailty, sarcopenia, and locomotive syndrome, which develops due to a lack of exercise, have been considered lifestyle-related diseases as well [110]. Not only the excess consumption of foods with high calories but also the decline in metabolism as the result of a lack of exercise is responsible for the manifestation of metabolic syndrome. A report shows that 1–2 days a week of walking over 8000 steps avoids this lack of exercise and reduces the risk of death [111]. In the case of Japan, where the population is ageing rapidly, lifestyle-related diseases such as cancer, cardiovascular diseases, diabetes, and chronic obstructive pulmonary disease (COPD) are a serious problem, as they are responsible for about 30% of medical expenses and 60% of deaths. On the other hand, some people are interested in understanding their own health condition and hoping to improve their lifestyle, especially regarding metabolic syndrome [112]. The term “lifestyle-related diseases” is defined as a group of diseases whose onset and progression are mainly related to lifestyle habits such as eating, exercise, rest, smoking, and drinking. Diet-associated lifestyle-related diseases include diabetes, obesity, hyperlipidaemia, hyperuricemia, cardiovascular diseases, colon cancer, and periodontal disease. Exercise-related diseases include diabetes, obesity, hyperlipidaemia, and hypertension. In addition, smoking is associated with lung cancer, cardiovascular disease, COPD, and periodontal disease, and alcohol consumption is associated with alcoholic liver disease [113].

Improvement in lifestyle-related disease conditions requires the improvement of diet and engaging in exercise [114]. These disorders are related to metabolic syndrome induced by the accumulation of excess energy in the body. Disorders related to metabolic syndrome are induced by chronic inflammation through the production of the pathogenic contributor adipokine TNF-α. A massive amount of TNF-α production suppresses GLUT4 production and leads to the development of insulin resistance, causing diabetes, atherosclerosis, liver failure, and dementia. On the other hand, the protector chemical adiponectin induces the production of insulin from pancreatic Langerhans β-cells and suppresses the onset of metabolic-syndrome-related diseases [115]. A randomised controlled trial revealed that the intake of polyphenol in a Mediterranean diet halved the onset of non-fatty liver disease (NFALD) associated with metabolic syndrome [116].

**Table 3 ijms-24-12167-t003:** Health-promoting effects of phytochemicals and vitamin D against disorders.

Diseases		Phytochemicals/ Vitamin D	Main Findings and Markers	Dose	Subjects	First Author, Year [Ref.]
Lifestyle- related diseases	Metabolic syndrome/ anti-obese	Quercetin	Visceral fat area (VFA) in low HLD subjects ↓	9 g/day: 12 w	Human (CRT)	Nishimura, M., 2019 [19]
		Hesperidin	Blood glucose ↓, Liver weight ↓, NAFLD ↓, NO ↓, IL-6 ↓, TNF-α ↓	50 mg–10 g/kg: 70 min–4 w	Mouse/rat	Xiong, H., 2019 [22]
			PPAR-γ ↓, C/EBPβ ↓, SREBP1 ↓, ROS ↓, ACDC ↑, IL-6 ↓, TNF-α ↓, NO ↓	0.1–50 μM: 1 min-8 d	In vitro	
		Nobiletin	Serum amylase ↓, Pancreatic myeloperoxidase activity ↓, Inflammatory factors ↓, p-p38 ↓, AKT ↓	50 mg/kg	Mouse	Chagas, MDSS, 2022 [24]
		Sudachitin	Bmal1 ↑, Liver triglyceride ↓, TGF-β, TNF-α ↓	50 or 100 mg/kg	Rat	Mawatari, K., 2023 [27]
		Tea catechins	Body weight ↓, BMI ↓, Blood LDL/HDL ratio ↓	EGCG 146 mg + hesperidin 178 mg/day: 12 w	Human (RCT)	Yoshitomi, R., 2021 [28]
			Glycerol ↓	2.3, 11.5 μM	In vitro	Chen, S., 2015 [31]
			Liver β oxidation activity ↓	0.1–0.5% (*w*/*w*)	Mouse	Murase, T., 2002 [32]
		Cocoa flavanols	Insulin ↑, Moderate low blood sugar level	Chocolate bar 20–100 g/day containing 15–500 mg polyphenol	Human	Strat, K.M., 2016 [39]
			Prebiotics effects for gut microbiota ↑, Gut barrier function ↑, Endotoxin absorption ↓	Addition of cocoa powder 0.5–10% (*v*/*v*) in diet (mouse/rat)	Mouse/rat	
		Vitamin D	Osteoporosis ↓	Subject with serum 25-hydroxyvitamin D < 25 nM (10 ng/mL) needs more trial	Human	Gallagher, J.C., 2023 [82]
	Diabetes	Naringin	Serum IL-6 ↓	Mediterranean Diet Intervention: 12 w	Human (CRT)	Al-Aubaidy HA, 2021 [23]
		Anthocyanins	Myocardial infarction risk ↓, Diabetes risk ↓, Mortality of cardiovascular diseases ↓	Daily intake of blueberry or anthocyanins 25–500 mg/day	Human	Kalt, W., 2020 [43]
			Retinal inflammation ↓	Bilberry extract 500 mg/kg/day: 4 d	Mouse	
		Grape polyphenols (in grape pomace (GP))	*Prevotella ↓, Firmicutes* ↓ miR-222 ↑ in responder subjects.	8 g/day: 6 w	Human (RCT)	Ramos-Romero, S., 2021 [78]
	Blood sugar spikes	Cacao polyphenols	Insulin ↑, Serum GLP-1 ↑	635 mg/day	Human (RCT)	Kawakami, Y., 2021 [40]
		Grape polyphenols (in grape juice)	Hunger ↓, Appetite ↓	355 mL/day: 8 w	Human (RCT)	Coelho, O.G.L., 2021 [71]
	Cardiovascular disease	Resveratrol	LDL ↓, Triglyceride ↓	250–1000 mg/kg/day	Human	Bonnefont-Rousselot, D., 2016 [41]
			miRNA expression ↑	5 mg/kg/day: 21 d	Rat	
		Rutin	Carbohydrate absorption in small intestine ↓, Glucose generation in the tissue ↓, Tissue glucose incorporation ↑, insulin secretion ↑	50 or 100 mg/kg	Rat	Ghorbani, A., 2017 [26]
Lifestyle- related diseases	Cardiovascular disease	Grape polyphenols (in GP)	Blood pressure ↓ Fasting blood glucose ↓	2 g/day: 6 w	Human (RCT)	Taladrid, D., 2022 [68]
		Grape polyphenols (in black seed raisin)	Diastolic blood pressure (DBP) ↓ Serum total antioxidant capacity (TAC) ↑	90 g/day: 5 w	Human (RCT)	Shishehbor, F., 2022 [69]
		Grape polyphenols (in GSE)	Blood pressure ↓	300 mg/day: 16 w	Human (RCT)	Schön, C., 2021 [47]
	Atherosclerosis	Quercetin	Serum quercetin-3-glucuronide (Q3GA) ↑, Cardiovascular disease risk ↓	350–500 g of cooked onion paste roasted with salad oil.	Human	Kawai, Y., 2008 [42]
			Q3GA accumulation in macrophages ↑, Form cell formation ↓	1 μM	In vitro	
	Liver failure	Curcumin	Lung fibrosis ↓, NF-κΒ ↓	1500 mg/day: 12 w	Human (RCT)	Saadati, S., 2019 [38]
	Dementia	Hesperidin	Cognitive function ↑, Executing function ↑, Episodic memory ↑	32 or 275 mg/day: 8 w	Human	Hajialyani, M., 2019 [21]
		Nobiletin	AD pathology ↑, Motor function ↑, Cognitive function ↑, Aβ ↓, Tau hyperphosphorylation ↓	10–50 mg/kg, i.p. or p.o.	Mouse/rat	Nakajima, A., 2019 [25]
		Catechins	Cognitive impairment ↓	Green tea intake 1–6 cups/week (systematic review)	Human	Kakutani, S., 2019 [33]
	Anti-ageing	Procyanidins	Physical dysfunction ↓, Pathophysiology ↓, Survival of aged mice ↑	PCC1 20 mg/kg i.p.	Mouse	Xu, Q., 2021 [44]
			Cell viability ↑, Apoptosis in senescent cells ↑, BCL-2 ↓, Caspase 3, 9 ↑	PCC1 100 μM	In vitro	
Intestinal disorders		Oleanolic acid	Large intestine contraction ↑	1–100 μM (measurement in mouse tissue)	Mouse	Alemi, F., 2013 [46]
		Flavonoids	*Faecalibacterium prausnitzii* ↑, *Ruminococcaceae* ↑, *Klebsiella* spp. ↓, *Prevotella* spp. ↓, F/B ratio ↓	Dried fruits (prunes) approximate 100 g/day	Human	Alasalvar, C., 2023 [117]
			TNF-α ↓, Leukocyte attachment ↓, micromolecular permeability ↓, Τissue injury ↓	Micronised flavonoid fraction (Daflon) 500 mg (mouse)	Mouse	Kumazawa, Y., 2006 [118]
			TNF-α ↓, Cell membrane LPS-induced raft accumulation ↓	200 μM	In vitro	
		Raisin phytochemicals	*Faecalibacterium prausnitzii* ↑, *Bacteroidetes* spp. ↑, *Ruminococcus* spp. ↑	Dried raisins 85 g/day: 2 w	Human	Wijayabahu, A.T., 2019 [119]
		Grape polyphenols (in grape powder)	Gut microbiota α diversity index ↑ *Verrucomicrobia* ↑ *Akkermansia* ↑	46 g/day: 4 w	Human (RCT)	Yang, J., 2021 [72]
		Vitamin D	Bone metabolism ↑, risk of falls ↓	25(OH)D3 between 20 and 125 nM to obtain the certain skeletal effects without toxic effects	Human	Sassi, F., 2018 [120]
Intestinal disorders		Vitamin D	VitD deficiency induces VDR ↑, VDBP ↑, P450 CYP27b1 ↑, Th1 ↑, Th2 ↑, Th17 ↑, Treg ↓	Normal: 1000, Deficiency: >25 IU VitD3/kg	Mouse	Huang, F., 2020 [121]
Lung diseases	COVID-19	EGCG	Viral infectious ability (TCID_50_) ↓, Viral RNA reproduction ↓, Second viral generation ↓	1 mM, 40 μM, 60 μM	In vitro	Ohgitani, E., 2021 [29]
			Viral infectious ability (TCID_50_) ↓, Second viral generation ↓	Tea catechins in saliva	In vitro	Ohgitani, E., 2021 [30]
		Vitamin D	LL-37/leukocyte count ratio as a hospital admission indicator	Mean serum calcitriol levels (active vitamin D hormone) were within the reference range of 20–79 ng/L	Human	Keutmann, M., 2022 [122]
		Phytochemicals and vitamin D	Perturb viral cellular infection by vitamin D, Anti-oxidative effects ↑, Anti-inflammation ↑	Meta analysis showed improvement of COPD	Human	Iddir, M., 2020 [84]
	Lung fibrosis	Quercetin	Fibrosis inducing p16 expressed senescent fibroblast ↓	Quercetin 50 mg/kg + Dasatinib 5 mg/kg p.o.: 3 times	Mouse	Schafer, M.J., 2017 [123]
		Quercetin and vitamin D	TNF-α ↓	Sufficient serum vitamin D level is over 30 ng/mL	Human	Santa, K., 2023 [12]
			TNF-α ↓	Quercetin 5 μΜ	In vitro	
		K-FGF	Serum IgE ↓, Neutrophil numbers ↓, PCA reaction ↓	100 mg/kg/day: 17 d	Mouse	Tominaga, T., 2010 [49]
		K-FGF	Th1/Th2 balance ↑, Antigen specific IgE production ↓	450 or 675 mg/day	Human	Kumazawa, Y., 2014 [50]
Sepsis		Hesperidin	TNF-α ↓, Serum LPS ↓	1 mg	Mouse	Kawaguchi, K., 2004 [124]
			Caspase 3 ↓, BCL-2 ↓, TLR4 ↓, HSP70 ↓, MyD88 ↓, TNF-α ↓, IL-6 ↓	Hesperidin 10–20 mg/kg i.v.	Mouse	Mahomoodally, M.F., 2022 [125]
		Naringin	TNF-α ↓, IL-6 ↓, NF-κΒ ↓, iNOS ↓, HO-1 ↑	Naringin 50–100 mg/kg p.o.		
		Quercetin	TNF-α ↓, IL-1β ↓, HMGB1 ↓, iNOS ↓, NO production ↓	Quercetin 1–100 mg/kg i.p.		
		Resveratrol	TNF-α ↓, MMP-9 ↓, IL-6 ↓, iNOS ↓, NLRP3 ↓, E-selectin/ICAM- -1 ↓, SIRT-1 ↑, IL-10 ↑	Resveratrol 1–60 mg/kg i.p.		
		Curcumin	IL-1β ↓, IL-6 ↓, TNF-α ↓, Caspase ↓, SMAD3 ↓, PPAR-γ ↑	Curcumin 50–200 mg/kg i.p.	Rat	
		Cyanidin	IL-1β ↓, IL-6 ↓, TNF-α ↓, PG ↓, Anti-oxidative effect ↑	Cyanidins 10–30 mg/kg i.p.		
		Silymarin	IL-1β ↓, PGE2 ↓, IL-6 ↓, TNF-α ↓	Silymarin 50–100 mg/kg i.p.		
		Vitamin D	Correlation of vitamin D and LL-37 levels	Placebo, 200,000 IU, 400,000 IU: 24 h within severe sepsis or septic shock	Human (RCT)	Quraishi, S.A., 2023 [126]

RCT = Randomised clinical trial. ↑: increase. ↓: decrease.

### 4.2. Intestinal Disorders and Gut Microbiota

Intestinal disease is a particular problem among the young generation, especially irritable bowel syndrome (IBS) and inflammatory bowel disease (IBD). IBD is an autoimmune disease divided into two types, ulcerative colitis (UC) and Crohn’s disease. UC was designated as an intractable disease in Japan in the 1970s and was a rare disorder. However, the number of patients exceeded over 10,000 in 1985 and reached over 77,000 in 2002, especially in the younger generation in their twenties. In addition, Crohn’s disease, which manifests in the small intestine as well as large intestine, has significantly increased in the same generation [127]. The underlying causes of these disorders include a westernisation of the diet, with excess amounts of animal protein and fat, which tend to induce dysbiosis. Furthermore, a stressful lifestyle is another cause of these diseases [128].

A good example of the health benefit of the diversity of microbiota has been shown with healthy athletes having 22 more microbiota than did other groups of people [129]. There are two dominant gut microbiota: *Firmicutes* and *Bacteroidetes*. *Firmicutes* are Gram-positive bacteria commonly known as obese-type bacteria which tend to increase in the presence of high-fat and high-sugar diets and massive meat consumption and are responsible for metabolic syndrome. On the other hand, *Bacteroidetes* are Gram-negative bacteria known as lean-type bacteria. Keeping the F/B ratio low is considered healthier [130]. Figure 3 indicates that the influence of phytochemicals and vitamin D is beneficial for lowering the F/B ratio among gut microbiota. Both phytochemicals and vitamin D are beneficial in lowering the F/B ratio. To maintain healthy microbiota, 3PDs, including prebiotics, probiotics, phytochemicals, and vitamin D, are important. Prebiotics include dietary fibre, especially microbiota-accessible carbohydrates (MACs), and water-soluble fibres, which are eaten by health-beneficial gut microbiota [131]. Probiotics are healthy gut microbiota, including lactic acid bacteria, *Bifidobacterium*, and butyric-acid-producing bacteria, usually contained in fermented foods such as yogurt. They foster healthy gut microbiota and suppress metabolic syndrome and related disorders [132]. Butyric acid produced by the butyric-acid-producing bacteria *Clostridium butyricum* and *Faecalibacterium prausnitzii* induces regulatory T cells (Treg) in the gut, and IL-10 produced by Tregs helps suppress inflammatory responses and upregulate immunity [133]. Furthermore, flavonoid phytochemicals work as prebiotics and upregulate immunity by increasing beneficial microbiota such as butyric-acid-producing bacteria [117] and suppress chronic inflammation through lowering the production of TNF-α [118]. In addition, it is intriguing that the consumption of raisins is reported to induce beneficial changes in gut microbiota [119]. Here, vitamin D is also important for maintaining healthy gut microbiota and the activation of Tregs [120,121].

### 4.3. COVID-19 and Lung Disorders

In addition to COVID-19, tuberculosis, and pneumococcal infection, typical lung diseases include asthma, sleep apnoea syndrome, interstitial pneumonia, lung cancer, and COPD. COVID-19 is caused by SARS-CoV-2 virus infection, with lung cells expressing the ACE2 receptor [134]. Infected cells undergo cell death after viral proliferation, release damage-associated molecular patterns (DAMPs) and viral particles, and induce inflammation. COVID-19 patients experience cytokine storms in the lung and disseminated intravascular coagulation (DIC) and, in the worst case, fatal dyspnoea [135]. In particular, the conditions of elderly persons and patients with underlying conditions are more likely to become serious and need critical caution. Viral infection causes various side effects, not only in the lungs, but also in other organs [136]. Bacterial infections such as tuberculosis and pneumococcus induce pneumonia and are generally classified into community-acquired pneumonia and hospital-acquired pneumonia (HAP) [137]. Recently, new types of infectious pneumonia such as *Mycobacterium avium* complex (MAC) are emerging [138]. Typical asthma is induced by allergic responses [139]; cough-variant asthma is the result of more than 8 weeks of continuous coughing after a cold that eventually turns into asthma [140]. Sleep apnoea syndrome onsets with airway obstruction whilst sleeping, mainly because of obesity [141]. Interstitial pneumonia (IP) is pulmonary fibrosis in lung alveoli and causes dyspnoea because of difficulties in oxygen incorporation into the body due to alveolar wall fibrosis. In addition to COPD, IP often develops as a sequela of COVID-19 because fibroblasts recruited by TGF-β released from M2 macrophages induce lung fibrosis [142]. Cancer, including lung cancer, is the most fatal disease in developed countries. Over 60% of cancer onsets are due to lifestyle-related habits, including smoking and alcohol consumption, and statistically, 20% of cancer develops from abnormal cell division in relation to inflammation [143]. COPD is the so-called tobacco disease in Japan because over 90% of COPD in Japan is correlated with smoking. Both emphysema and chronic bronchitis are combined in COPD. COPD-induced dyspnoea causes various systemic disorders including ischemic heart disease, osteoporosis, diabetes, and metabolic syndrome [144].

Figure 4 is the molecular mechanism of TNF-α suppression in macrophages by the intake of flavonoid quercetin and vitamin D [12]. Fermented grape foods from Koshu, a Japanese grape strain (K-FGF), contain the grape skin and seed paste of *Vitis vinifera* Koshu, fermented with vegetable lactic acid bacteria. K-FGF suppresses chronic-inflammation-related disorders though the suppression of TNF-α [49,50]. The combined effects of quercetin and vitamin D suppress TNF-α production. In pulmonary fibrosis, TGF-β produced from M2 macrophages recruits fibroblasts to the lung lesion and accumulates collagen to induce fibrosis. Some intriguing research has shown that flavonoids suppress lung fibrosis [123].

### 4.4. Sepsis and Infectious Diseases

Patients with underlying conditions and lifestyle-related diseases are more likely to manifest sepsis in addition to coronavirus infection because of its onset mechanism [145]. All kinds of infectious diseases, such as pneumonia, urinary tract infection, and peritonitis, induce sepsis. Bacterial infection is the most notorious cause of sepsis, but every microbe, including bacteria, viruses, and fungi, can be responsible. Once manifested, patients with sepsis experience organ failure and uncontrollable biological reactions. Septic shock is the worst form of sepsis, with acute inflammation and severe organ failure as the result of infection; it is associated with severe circulatory and cellular metabolic abnormalities [146]. The condition called cytokine storm is induced by a runaway immune response accompanied by the excess production of pro-inflammatory cytokines, including TNF-α, IL-1β, and IL-6 [147,148]. Macrophages are already activated in underlying conditions, lifestyle-related diseases, and metabolic syndrome, and they produce large amounts of TNF-α when stimulated by DAMPs (mainly mitochondrial DNA, which damages cells and results in debris) as well as virus particles. As a result, the activation of the coagulation system induces thrombi and decreased blood circulation in organs, which causes organ failure, which in the worst case, leads to death. Furthermore, anti-inflammatory cytokine treatment may not effectively cure patients with sepsis [149,150]. When septic shock is observed, large amounts of transfusion and oxygen inhalation are conducted as a treatment. Blockade of TNF-α production with steroids, immunosuppressive drugs, and biological medical products (anti-TNF-α antibodies, etc.) is effective; however, delays in treatment significantly reduce the chance of survival in sepsis. According to clinical practice guidelines, after evaluation indicating the potential of sepsis, intervention needs to be performed within 3 h, indicating that rapid treatment is important [151].

Recently, the antimicrobial peptide LL-37 has been given attention for the suppression of sepsis. LL-37 is the only human cathelicidin, an antimicrobial peptide composed of 37 amino acids expressed mainly in neutrophils and epithelial cells suppressing sepsis through activating natural immunity. LL-37 released from neutrophils suppresses sepsis in mice [152] and inhibits IL-1β and TNF-α production in macrophages by suppressing LPS-stimulated CD14/TLR4 activation and the subsequent NF-κB signalling pathway and activation of P2X7 by ATP from dead/dying cells [153]. Figure 5 shows the molecular mechanism of inflammatory cytokine suppression by LL-37. LL-37 suppresses the binding of LPS to CD14/TLR4 on target cells and suppresses the activation of Caspase-1, IL-1β expression, and pyrosis-induced cell death in vitro [154]. Furthermore, bacterial infection is also involved in the development of atherosclerosis. In this pathway, LL-37 highly activates NF-κB p65 in LPS-induced senescent vascular endothelial cells, when compared with non-senescent cells, suggesting the importance of LL-37 in the TLR4-LPS pathway [155]. In the 1920s, patients with pulmonary tuberculosis received treatment with regular sun exposure before the invention of antibiotics. This treatment involved the enhancement of LL-37 by vitamin D to upregulate natural immunity [156]. A randomised trial showed that patients with septic shock and severe sepsis rapidly induced LL-37 expression after vitamin D supplementation [126]. According to a report, vitamin D supplementation induced LL-37 in children with atopic dermatitis [157]. Furthermore, vitamin D supplementation enhanced the expression of LL-37 and increased the resistance to SARS-CoV-2 viral infection in COVID-19 patients [122].

Septic shock is common in neonates, elderly people, and pregnant women. Leukopenia induced by antibiotics and corticosteroid use and the use of invasive devices (endotracheal and endotracheal tubes, drains, etc.) in the hospital may trigger sepsis [158]. Furthermore, patients with cancer or lifestyle-related diseases, including diabetes and liver cirrhosis, tend to experience septic shock. In addition, immunocompromised patients using immunosuppressive agents are reported to easily experience sepsis [159]. Previous research has shown that vitamin C prevents bacterial or viral infections and that a lack of vitamin A leads to susceptibility to the infection of viruses [160,161]. Lactoferrin enhances natural immunity and is effective in the suppression of inflammatory responses induced by viral infections [162]. The polyphenol hesperidin has been shown to have protective effects against sepsis [124]. Other anti-inflammatory phytochemicals such as hesperidin, naringin, quercetin, resveratrol, and silymarin have been shown to be effective in sepsis management [125,163]. Maintaining a healthy state and avoiding ME-BYO and lifestyle-related diseases through a healthy diet with the intake of phytochemicals combined with vitamin D makes it possible to reduce the risk of sepsis via the suppression of proinflammatory cytokine production.

## 5. Conclusions

Phytocompounds have been widely used in many places, such as in traditional medicine, of which *Kampo* is a good example. In addition, the phytochemicals contained in fruits and vegetables have been recognised for their health-promoting effects, mainly their anti-inflammatory effects against chronic inflammation. Recently, phytochemicals have been recognised in our country as the seventh nutrient. In addition, the combined intake of vitamin D with phytochemicals upregulates immune responses. Furthermore, the idea of ME-BYO, finding the non-disease condition and keeping the healthy condition, has been spreading in Japan. There is a growing movement to preserve the healthy condition by preventing underlying conditions, lifestyle-related diseases, metabolic syndrome, and obesity by incorporating a healthy diet and exercise. This is very important for the prevention of various disorders triggered by lifestyle-related diseases, intestinal diseases commonly found in younger generations, and sepsis, induced from infectious disease, accompanied with multiple organ failure. Taking countermeasures against ME-BYO and lifestyle-related diseases is necessary because they increase medical expenses and increase the incidence of disorders with high mortality rates. The intake of phytochemicals and vitamin D enhances anti-inflammatory effects, upregulates immunity, and prevents chronic disorders through the maintenance of gut microbiota. Evidence acquired during the COVID-19 pandemic revealed that having underlying conditions such as lifestyle-related diseases induced by chronic inflammation increases the risk of infectious disease. Therefore, daily improvement and prevention measures for these underlying conditions are very important. Our research predicts the necessity of the intake of phytochemicals and vitamin D for a healthy life, and the prevention of ME-BYO and a variety of diseases in the near future.

## Figures and Tables

**Figure 1 ijms-24-12167-f001:**
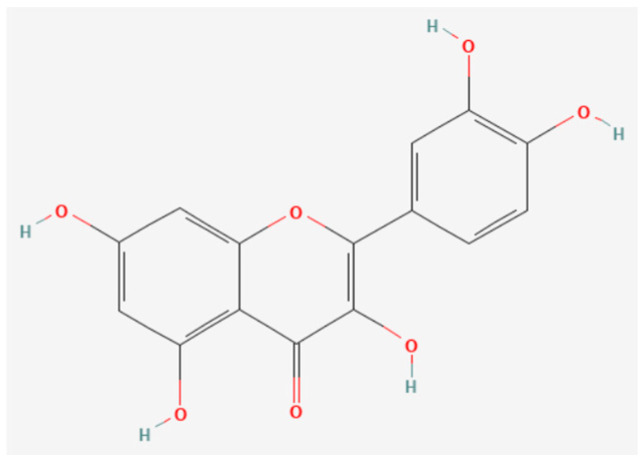
Chemical structure of quercetin, a typical phytochemical. (Downloaded from PubChem CID: 5280343.).

**Figure 2 ijms-24-12167-f002:**
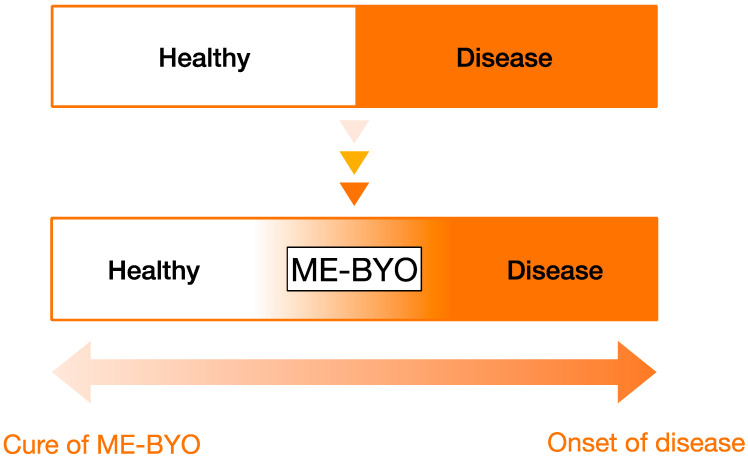
Schematic view of ME-BYO. We cannot distinguish between a healthy state and disease. ME-BYO (non-disease state) is the state continuously changing from healthy state to disease. Generally, ME-BYO is considered to be a state in which there are no subjective symptoms but abnormalities in examinations, or a state in which there are subjective symptoms, but no abnormalities in examinations. The importance of ME-BYO has been increasing because the prevention of ME-BYO in the younger generation will reduce medical expenses and prolong healthy longevity.

**Figure 3 ijms-24-12167-f003:**
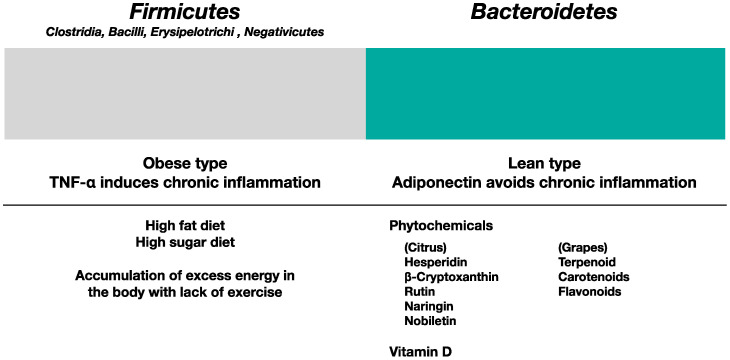
The influences of phytochemicals and vitamin D in gut microbiota. High-fat, high-sugar and high-calorie diet causes obesity; without sufficient exercise, people tend to the onset of metabolic syndrome. At this stage, gut microbiota tends to be of the *Firmicutes* dominant, obese-prone type and induce chronic inflammation by the production of TNF-α. On the other hand, the intake of prebiotics, probiotics, polyphenol, and vitamin D (3PDs) induces increasing *Bacteroidetes* dominant healthy microbiota with associated adiponectin production. An increase of *Bacteroidetes* bacteria lowers the F/B ratio in the gut [62].

**Figure 4 ijms-24-12167-f004:**
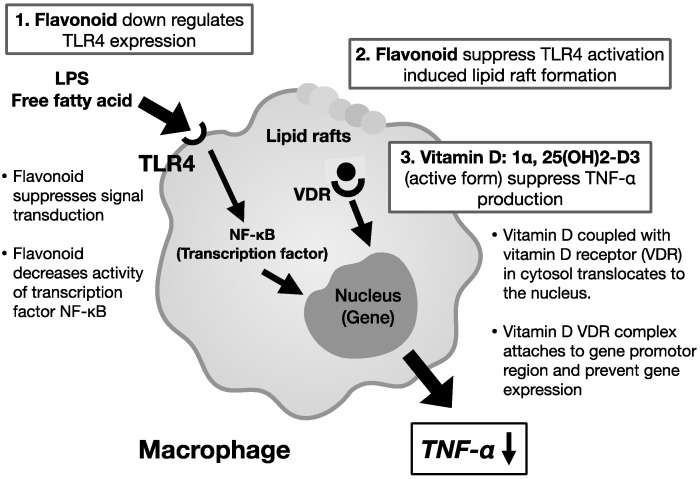
TNF-α suppression by flavonoids and vitamin D. Inflammatory cytokine TNF-α induces chronic inflammation. However, the flavonoid quercetin suppresses the TNF-α production through the suppression of TLR4 by LPS and free fatty acid. In addition, quercetin suppresses lipid raft formation on the surface of the macrophage as well as suppressing TLR4 activation. Then, activated vitamin D composes vitamin D-VDR complex in cytosol and moves into the nuclei, where it precludes TNF-α gene expression. These mechanisms suppress the TNF-α production and ameliorate chronic inflammation [12].

**Figure 5 ijms-24-12167-f005:**
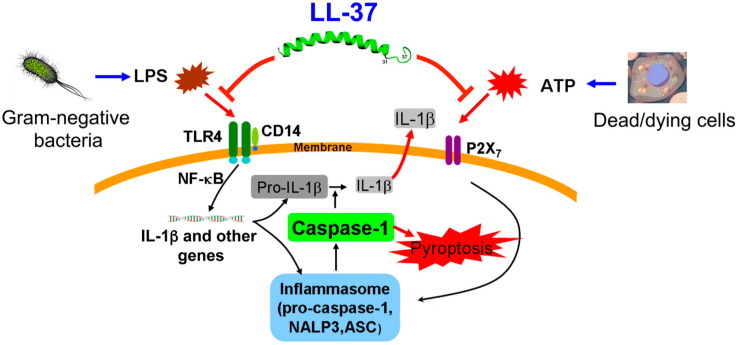
Suppression of LPS/ATP-induced macrophage cell death in sepsis by LL-37. Gram-negative bacterial LPS and dead/dying-cell-derived ATP induce macrophage cell death (pyroptosis) via the action on CD14/TLR4 and P2X7, respectively. LL-37 reduces the LPS/ATP-stimulated pyroptosis of macrophages and IL-1β production by both curtailing the action of LPS on CD14/TLR4 and blocking the P2X7 response to ATP [13].

**Table 1 ijms-24-12167-t001:** Health promoting effects of phytochemicals in fruits and vegetables.

Fruits/ Vegetables	Phytochemicals	Effects	Main Findings and Markers	Dose	Subjects	First Author, Year [Ref.]
Onion	Quercetin	Obesity prevention, liver function improvement	Visceral fat area (VFA) in low HLD subjects ↓	9 g/day: 12 w	Human (CRT)	Nishimura, M., 2019 [19]
	Rutin (Quercetin- glycoside)	Increasing cell viability	Cell viability ↑, G2, M phase cells ↑, IEGs ↑, iPSCs ↑	0.05–100 μM	In vitro	Miyake, T., 2021 [20]
Citrus fruits	Hesperidin	Neuroprotective effects	Cognitive function ↑, Executing function ↑, Episodic memory ↑	32 or 275 mg/day: 8 w	Human	Hajialyani, M., 2019 [21]
		Lowering cholesterol and blood pressure, maintaining bone density, blood vessel protection in sepsis	Blood glucose ↓, Liver weight ↓, NAFLD ↓, NO ↓, IL-6 ↓, TNF-α ↓	50 mg-10 g/kg: 70 min-4 w	Mouse/rat	Xiong, H., 2019 [22]
			PPAR-γ ↓, C/EBPβ ↓, SREBP1 ↓, ROS ↓, ACDC ↑, IL-6 ↓, TNF-α ↓, NO ↓	0.1–50 μM 1 min-8 d	In vitro	
	Naringin	Reduction of inflammation markers in diabetic patients	Serum IL-6↓	Mediterranean Diet Intervention: 12 w	Human (CRT)	Al-Aubaidy HA, 2021 [23]
	Nobiletin	Anti-oxidative, skin metabolism, blood sugar modulation by adiponectin production, anti-dementia	Serum amylase ↓, Pancreatic myeloperoxidase activity ↓, Inflammatory factors ↓, p-p38 ↓, AKT ↓	50 mg/kg	Mouse	Chagas, MDSS, 2022 [24]
			AD pathology ↑, Motor function ↑, Cognitive function ↑, Aβ ↓, Tau hyperphosphorylation ↓	10–50 mg/kg, i.p. or p.o.	Mouse/rat	Nakajima, A., 2019 [25]
	Rutin (Quercetin- glycoside)	Blood vessel protection/anti-diabetes	Carbohydrate absorption in small intestine ↓, Glucose generation in the tissue ↓, Tissue glucose incorporation ↑, insulin secretion ↑	50 or 100 mg/kg	Rat	Ghorbani, A. 2017 [26]
	Sudachitin	Prevention of liver fat through modulation of circadian clock	Bmal1 ↑, Liver triglyceride ↓, TGF-β ↓, TNF-α ↓	100 mg/kg: 22 w	Mouse	Mawatari, K., 2023 [27]
Tea	Catechins	Anti-obesity effects with orange polyphenol	Body weight ↓, BMI ↓, Blood LDL/HDL ratio ↓	EGCG 146 mg + hesperidin 178 mg/day: 12 w	Human (CRT)	Yoshitomi, R., 2021 [28]
		Inactivation of SARS-CoV-2 virus	Viral infectious ability (TCID_50_) ↓, Viral RNA reproduction ↓, Second viral generation ↓	1 mM, 40 μM, 60 μM	In vitro	Ohgitani, E., 2021 [29]
			Viral infectious ability (TCID_50_) ↓, Second viral generation ↓	Tea catechins in saliva	In vitro	Ohgitani, E., 2021 [30]
		Lipolysis effect	Glycerol ↓	2.3, 11.5 μM	In vitro	Chen, S., 2015 [31]
		Visceral fat accumulation prevention	Liver β oxidation activity ↓	0.1–0.5% (*w*/*w*)	Mouse	Murase, T., 2002 [32]
		Prevention of dementia	Cognitive impairment ↓	Green tea intake 1–6 cups/week (systematic review)	Human	Kakutani, S., 2019 [33]
	Theaflavin	Antivirus, anti-inflammatory, anti-oxidative, anti-obesity	3CL-protease (constricting functional viral protein) ↓	8.44 μg/mL	In vitro	Shan, Z., 2021 [34]
Soybeans	Genistein Daidzein	Oestrogenic effects	Blood sugar ↓, Bone density in menopause women ↑, Breast cancer risk ↓	Genistein/Daidzein 200 mg/kg	Mouse	Nakai, S., 2019 [35]
	Equol	Oestrogenic gut microbial metabolite	Frequency of hot flushes during menopause ↓	10–30 mg/day: 8–12 w	Human	Mayo, B., 2019 [36]
		Metabolic syndrome prevention	Body weight ↓, BMI ↓, TC ↓, LDL-C ↓, non-HDL-C ↓, apoB ↓	Soy protein 30 g/day: 12 w (epidemiological research)	Human	Yamagata, K., 2021 [37]
			Blood sugar ↓, Glucose tolerance ↑, Blood insulin ↑	Genistein 20–250 mg/day: 8–12 w, Daidzein 50 mg/day: 24 w	Mouse	
			Lipid accumulation ↓, Lipid droplet accumulation ↓, C/EBPα ↓, PPAR-γ ↓, 2/FABP4 ↓	Culture with genistein 50–200 μM: 14 d	In vitro	
Turmeric	Curcumin	Liver protective effects	Lung fibrosis ↓, NF-κΒ ↓	1500 mg/day: 12 w	Human (RCT)	Saadati S., 2019 [38]
Cacao	Cacao polyphenols (Epicatechin, catechin, procyanidins)	Improvement in metabolic-syndrome-related disorders	Insulin↑, Moderate low blood sugar level	Chocolate bar 20–100 g/day containing 15–500 mg polyphenol	Human	Strat, K.M., 2016 [39]
			Prebiotics effects for gut microbiota ↑, Gut barrier function ↑, Endotoxin absorption ↓	Addition of cocoa powder 0.5–10% (*v*/*v*) in diet (mouse/rat)	Mouse/rat	
		Enhancement in GLP-1 and insulin production	Insulin ↑, Serum GLP-1 ↑	635 mg/day	Human (CRT)	Kawakami, Y., 2021 [40]
Grape	Resveratrol	Cardiovascular disease	LDL ↓, Triglyceride ↓	250–1000 mg/kg/day	Human	Bonnefont-Rousselot, D., 2016 [41]
			miRNA expression ↑	5 mg/kg/day: 21 d	Rat	
	Quercetin	Chronic inflammation, atherosclerosis	Serum quercetin-3-glucuronide (Q3GA) ↑, Cardiovascular disease risk ↓	350–500 g of cooked onion paste roasted with salad oil.	Human	Kawai, Y., 2008 [42]
			Q3GA accumulation in macrophages ↑, form cell formation ↓	1 μΜ	In vitro	
	Anthocyanins	Anti-inflammatory, lowering cardiovascular diseases, diabetes, and fatality by anti-oxidative effects	Myocardial infarction risk ↓, Diabetes risk ↓, Mortality of cardiovascular diseases ↓	Daily intake of blueberry or anthocyanins 25–500 mg/day	Human	Kalt, W., 2020 [43]
			Retinal inflammation ↓	Bilberry extract 500 mg/kg/day: 4 d	Mouse	
	Procyanidins	Anti-ageing	Physical dysfunction ↓, Pathophysiology ↓, Survival of aged mice↑	PCC1 20 mg/kg i.p.	Mouse	Xu, Q., 2021 [44]
			Cell viability ↑, Apoptosis in senescent cells ↑, BCL-2 ↓, Caspase 3, 9 ↑	PCC1 100 μM	In vitro	
	Procyanidins	Gut-brain axis	Endothelial function measured by flow-mediated dilation (FMD) ↑	Cocoa flavanols 710 mg/kg	Human	Osakabe, N., 2022 [45]
			Blood pressure ↓, eNOS ↓	10 mg/kg (body weight): repetitive	Rat	
	Oleanolic acid	Activating intestinal peristalsis	Large intestine contraction ↑	1–100 μM (measurement in mouse tissue)	Mouse	Alemi, F., 2013 [46]
	GSE	Suppression of high blood pressure	Blood pressure ↓	300 mg/day: 16 w	Human (CRT)	Schön, C., 2021 [47]
	GSPE	Maintaining normal blood pressure	Systolic blood pressures (SBP) ↓, Diastolic BP (DBP) ↓, stiffness parameter β ↑, incremental elastic modulus (Einc) ↑, Pulse wave velocity (PWV)↑	200, 400 mg/day: 12 w	Human (CRT)	Odai, T., 2019 [48]
	K-FGF	Suppression of inflammatory cytokine TNF-α	Serum IgE ↓, Neutrophil numbers ↓, PCA reaction↓	100 mg/kg/day: 17 d	Mouse	Tominaga, T., 2010 [49]
			Th1/Th2 balance ↑, Antigen specific IgE production↓	450 or 675 mg/day	Human	Kumazawa, Y., 2014 [50]

RCT = Randomised clinical trial. ↑: increase. ↓: decrease.

**Table 2 ijms-24-12167-t002:** Summary of randomised controlled trial using grape polyphenol (2021 to present).

First Author, Year [Ref.]	Research	Polyphenols	Treatment	Dose	Type of Subjects	Main Findings	Results
Nho, H., 2022 [65]	Endothelial function and endurance performance	Grape Seed Extract (GSE)	GSE supplementation during cycling exercise	300 mg/day: 14 d	Athletes (*n* = 12)	VO_2_ peak ↓ Time to exhaustion ↑	Chronic supplementation of GSE improved endurance performance.
Buerkli, S., 2022 [66]	Iron absorption in adults with hereditary hemochromatosis (HH)	12 natural polyphenol supplementations, including grapes.	Fractional iron absorption (FIA) after polyphenol supplementation (PPS)	FeSO_4_ 10 g + PPS 2 g/day: 45 d	HH patients (*n* = 14)	FIA ↓	Reduced iron accumulation and frequency of phlebotomy in HH patients.
Van Doren, W.W., 2022 [67]	Serum polyphenol concentration and cognitive performance	Grape juice	Concord grape juice consumption	85 g: 0–2 w; 170 g: 3–4 w; 255 g: 4–24 w	Gulf War illness veterans (*n* = 26)	Serum polyphenol ↑ Cognitive performance ↑	Increased bioavailability of polyphenols and improved cognitive performance.
Taladrid, D., 2022 [68]	Interplay between hypertension, blood sugar and gut microbiota	Grape pomace (GP)	Consumption of GP-derived seasoning	2 g/day: 6 w	High-cardiovascular risk subjects (*n* = 17) and healthy subjects (*n* = 12)	Blood pressure ↓ Fasting blood glucose ↓	Modification of cardiometabolic risk factors and gut microbiota.
Shishehbor, F., 2022 [69]	Cardiovascular risk factors and total antioxidant capacity	Raisin	Consumption of black seed raisin	90 g/day: 5 w	Hyperepidemic patients (*n* = 38)	Diastolic blood pressure (DBP) ↓ Serum total antioxidant capacity (TAC) ↑	Effects in cardiovascular risk factors and serum antioxidant capacity.
Bell, L., 2022 [70]	Cognitive function	Grape seed polyphenol extract (GSPE)	Cognitive tests after GSPE consumption	400 mg/day: 12 w	Healthy young adults, GSEP (*n* = 30) or placebo (*n* = 30)	No effective cognitive benefits revealed.	In contrast to older and cognitive compromised populations, no improvement of cognitive functions in healthy young adults.
Coelho, O.G.L., 2021 [71]	Effects of grape flavour and polyphenol in glycaemia, appetite, and cognitive function	Grape juice	Concord grape (*Vitis lambrusca*) juice (CGJ): polyphenol-free grape flavoured drink (LP), or LP with reduced flavour (LPF) compared	355 mL/day: 8 w	Adults with excess body weight (*n* = 34)	Hunger ↓, Appetite ↓ in CGJ and LP groups	Eight weeks’ intake of grape juice reduced fasting blood gulches.
Yang, J., 2021 [72]	Effects in gut microbiome	Grape powder	Grape powder consumption (equivalent of 2 servings of table grapes)	46 g/day: 4 w	Healthy subjects	Gut microbiota α-diversity index ↑ *Verrucomicrobia* ↑ *Akkermansia* ↑	Significant changes in gut microbiota and cholesterol/bile acid metabolism.
Tutino, V., 2021 [73]	Impact in gastrointestinal cancer-related pathways, related circulating microRNA	Fresh table grape	Consumption of fresh grapes (Autumn Royal table grape)	5 g/day: 3 w	Healthy subjects (*n* = 40)	18 miRNAs ↓ 2 miRNAs ↑	Effects in miRNAs levels in counteracting cancer development, including gastrointestinal cancers.
García-Díez, E., 2021 [74]	Influence in postprandial metabolism	Grape powder	Single application of grape powder	46 g/day: Once	Obese subjects (30 ≤ BMI < 40) (*n* = 25)	Blood glucose, Insulin, Triglycerides, Uris acid, Blood count, Haemoglobin, Viscosity, Antioxidant capacity, and Satiety perception after 5 h.	Single supplementation showed no significant changes; insufficient amounts.
Das, M., 2021 [75]	Clinical efficacy in periodontal pockets	GSE	GSE injection to periodontal pockets	4% GSE in PBS: 12 w	Patients with periodontal pockets (*n* = 64)	Probing depth (PD) ↓ Relative attachment level (RAL) ↓	Beneficial in management of periodontal pockets.
Schön, C., 2021 [47]	Positively modulating blood pressure and perceived stress	GSE	Administration of GSE tablets (Envovita (GSEe))	300 mg/day: 16 w	Healthy subjects	Blood pressure ↓	Improvement of endothelial functionality in vitro, blood pressure, and positive effects in stress perception.
Dani, C., 2021 [76]	Oxidative stress, inflammation, and epigenetic modulation	Grape juice	Red grape (*Vitis labrusca*) juice group (GJG), GJG with exercise (GJEG), and placebo with exercise (PLEG) compared	400 mL/day: 4 w	Healthy elderly women aged 59 years and over (*n* = 29)	IL-6 ↓ in GJEG and PLEG	Physical training affects anti-oxidative and anti-inflammatory effects in elderly women, grape juice increased non enzymatic antioxidant defence.
Vors, C., 2021 [77]	Differences in cardiovascular diseases by sex	Combined polyphenol and L-citrulline	Administration of polyphenol plus L-citrulline	Polyphenol 548 mg + L-citrulline 2 g/day: 6 w	Men and women with prehypertension (*n* = 73)	DBP ↓, AGEs ↓ in women	Decrease of ambulatory systolic BP (SBP) in women. Sex-dependent BP response to polyphenol supplementation.
Ramos-Romero, S., 2021 [78]	Insulin’s response	GP	Supplementation of dried GP	8 g/day: 6 w	Subjects with at least 2 factors of metabolic syndrome (*n* = 49)	*Prevotella ↓,* *Firmicutes* ↓ miR-222 ↑ in responder subjects.	Faecal microbiota and miRNA expression are related, with variability in clinical trials with polyphenols.
Magrone, T., 2021 [79]	Improvement in nickel-mediated allergic contact dermatitis	Grape polyphenol	Administration of red grape (Nero di Troia cultivar) polyphenol	300 mg/day: 12 w	Allergic contact dermatitis (ADC) patients to Ni (*n* = 25)	IFN-γ ↓, IL-4 ↓, IL-17 ↓, pentraxin 3 ↓, NO ↓, IL-10 ↑	Anti-oxidative, anti-inflammatory, and anti-allergic properties of polyphenols were shown.

↑: increase. ↓: decrease.

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
