# Peer review of "Phytochemicals and Vitamin D for a Healthy Life and Prevention of Diseases"

_ijms, 2023, doi:10.3390/ijms241512167_

Round 1

Reviewer 1 Report (Previous Reviewer 1)

Thank you for the opportunity to review this manuscript. Covid 19 pandemic has rekindled the focus on the activity of natural substances in managing the clinical outcome. This review discusses the potential of phytochemicals with vitamin D to prevent disease and promote healthy lifestyles. This is a re-submission and the current version is greatly improved. Although it is a narrative review, I suggest to inlude a methodological section.

Author Response

Comments and Suggestions for Authors

Thank you for the opportunity to review this manuscript. Covid 19 pandemic has rekindled the focus on the activity of natural substances in managing the clinical outcome. This review discusses the potential of phytochemicals with vitamin D to prevent disease and promote healthy lifestyles. This is a re-submission and the current version is greatly improved. Although it is a narrative review, I suggest to inlude a methodological section.

Response: Thank you very much for reviewing our manuscript. We are very pleased to see your comment, “the current version is greatly improved”. In accordance with your comment, we added methodological section to the manuscript.

“(line-95) 2. Brief methodology of the review article

     “Briefly, we explain the methodology of the search method of scientific papers in the references of this review. This review is narrative review as we described above. We searched for papers in PubMed with combined keywords “phytochemicals”, “vitamin D” and/or the name of each phytochemical such as “polyphenol”, “quercetin”, and the name of disorders like “lifestyle-related diseases”, “gut microbiota”, “lung disease” and/or “sepsis”. After the search, we composed of this review article in reference to papers we considered necessary. Furthermore, over 50% of references are published within 5 years other than we thought important or well-known papers in the field. References include our previous publications and we have searched and collected scientific papers that are attractive especially in our region, including papers introduced in the press releases of universities in Japan, webpages from companies and institutions, and papers in the web news of featured special articles in related field from newspapers. References in this review include many “Review” papers as they summaries many research findings.

     In section 3, we summarised the experimental information that was conducted in the referenced papers regarding to phytochemicals contained in fruits and vegetables into table 1. We searched on PubMed for the keywords “grape and polyphenol” with article types “Clinical trial and Randomised Controlled Trial” for two years from 2021 to 2022 in addition to 2023 before this research. As a result, 16 search results were emerged and they are summarised in the table 2. Then, table 3 shows references organised by diseases in section 4 and benefits of phytochemicals and vitamin D consumption. Since table 3 contains some references show in table 1, the item “effects” was deleted, and they were re-sorted by “diseases”. Since there are many review articles in the reference papers, references that do not include detailed experimental methods or doses are excluded from the table.”

With this regard. the 3rd paragraph in “(line-245) 3.7 Grapes” has been changed from ...

     “Table 2 summarises the clinical trials related with grape polyphenols. We searched on PubMed for the keywords “grape and polyphenol” with article types “Clinical trial and Randomised Controlled Trial” for two years from 2021 to 2022 in addition to 2023 before this research, and 16 search results were emerged [62 - 76]. In addition to cardiovascular disorders, recently conducted variety of randomised controlled trials for metabolic syndrome related disorders including dementia, gut microbiota related research, periodontal diseases, allergic diseases, and cancer revealed intriguing findings. [63 - 77].” 

to ...

     “(line-284) Table 2 shows the clinical trials related with grape polyphenols. In addition to cardiovascular disorders, recently conducted variety of randomised controlled trials for metabolic syndrome related disorders including dementia, gut microbiota related research, periodontal diseases, allergic diseases, and cancer revealed intriguing findings [63 - 77].“.

Other Changes

Regarding to the tables, in accordance with the comment from another reviewer comment, the field “Main findings and markers” has been added to “Table 1 and 3” (please see in the manuscript). In “Table 2” there was the mix up of the units of measurement “ounce” and “gram (metric system)” and collected them together in “gram (metric system)”. New references have been added in the tables and text.

Changes with the addition of new references in tables and text:

Addition to the tables: (Please see in the manuscript)

Table 1 and 3.

Quercetin/Obesity prevention, liver function improvement (Table 1 only)/Visceral fat area (VFA) in low HLD subjects ↓ /9g/day: 12w/Human (RCT)/Nishimura M, 2019 [20]

Table 3.

Sepsis/Vitamin D/Correlation of vitamin D and LL-37 levels/Placebo, 200,000 IU, 400,000 IU: 24h within severe sepsis or septic shock/Human (RCT)/Quraishi SA, 2023 [153]

(We changed this reference because this reference was more suitable to the table.)

Moved within the table:

Table 3.

Dementia/Hesperidin/Cognitive function↑, Executing function↑, Episodic memory↑/32 or 275mg/day:  8w /Human/Hajialyani M, 2019 [25]

Dementia/Nobiletin/AD pathology↑, Motor function↑, Cognitive function↑, Aβ↓, Tau hyperphosphorylation↓/10-50mg/kg, i.p. or p.o./Mouse.rat/Nakajima A, 2019 [30]

Additional references:

20. Nishimura M.; Muro T.; Kobori M.; Nishihira J. Effect of Daily Ingestion of Quercetin-Rich Onion Powder for 12 Weeks on Visceral Fat: A Randomised, Double-Blind, Placebo-Controlled, Parallel-Group Study. Nutrients 2019, 12, 91.

140. Schafer MJ.; White TA.; Iijima K.; Haak AJ.; Ligresti G.; Atkinson EJ.; Oberg AL.; Birch J.; Salmonowicz H.; Zhu Y.; Mazula DL.; Brooks RW.; Fuhrmann-Stroissnigg H.; Pirtskhalava T.; Prakash YS.; Tchkonia T.; Robbins PD.; Aubry MC.; Passos JF.; Kirkland JL.; Tschumperlin DJ.; Kita H.; LeBrasseur NK. Cellular senescence mediates fibrotic pulmonary disease. Nat. Commun. 2017, 8, 14532.

144. Wautier JL.; Wautier MP. Pro- and Anti-Inflammatory Prostaglandins and Cytokines in Humans: A Mini Review. Int. J. Mol. Sci. 2023, 24, 9647.

153. Quraishi SA.; De Pascale G.; Needleman JS.; Nakazawa H.; Kaneki M.; Bajwa EK.; Camargo CA Jr.; Bhan I. Effect of Cholecalciferol Supplementation on Vitamin D Status and Cathelicidin Levels in Sepsis: A Randomized, Placebo-Controlled Trial. Crit. Care. Med. 2015, 43, 1928-1937.

Additional description to the text:

The new passage “(line-563) A randomised trial shows the patients with septic shock and sever sepsis rapidly induced LL-37 expression after the vitamin D supplementation [153].” has been added.

In addition, “(line-14) Abstract” have been modified. (Please see in the manuscript)

We sincerely hope our manuscript will be accepted and published in International Journal of Molecular Sciences, with your review.

We are looking forward to hearing from you at your earliest convenience.

Yours Sincerely,

Reviewer 2 Report (Previous Reviewer 2)

The authors have attempted to implement the tables, however, some aspects need special attention.

In detail:

- Units of measurement must be standardized (no Oz together with gram). The tables are intended to compare various data and with different units of measure and systems, this is not possible. I recommend using the metric system.

- It is not clear why so much data is missing. For example, if a reference is an in vitro study to evaluate a specific effect of a substance, the final concentration used should still be presented. It is not sufficient that the wording "Chemical analysis" is inserted. If no characterization suggests the effect, the reference should not be included. Similarly, it is not possible to evaluate an effect without an evaluation of the dose and therefore the frequent wording "no dose" makes me doubt whether the authors have inserted adequate references for the purpose or they may have insufficiently elaborated on the information present in the proposed references. If there is no dose because the study is observational, the range of concentrations for which an effect was noted relative to the comparator should be specified and entered.

- Some units of measurement suggest that the authors are unfamiliar with the quantity in question. μM/L is a meaningless unit of measurement. It should be μM or μmol/L.

- The topics of interest have been included in the tables but not enough outcomes and markers.

Author Response

Comments and Suggestions for Authors

The authors have attempted to implement the tables, however, some aspects need special attention.

In detail:

1. Units of measurement must be standardized (no Oz together with gram). The tables are intended to compare various data and with different units of measure and systems, this is not possible. I recommend using the metric system.

Response: Thank you very much for your review. We are very happy to see your careful and detailed comments. In accordance with your comment and recommendation, units described in “ounce” has been changed to “gram (metric system)” and all the units were standardised in metric system. 

2. It is not clear why so much data is missing. For example, if a reference is an in vitro study to evaluate a specific effect of a substance, the final concentration used should still be presented. It is not sufficient that the wording "Chemical analysis" is inserted. If no characterization suggests the effect, the reference should not be included. Similarly, it is not possible to evaluate an effect without an evaluation of the dose and therefore the frequent wording "no dose" makes me doubt whether the authors have inserted adequate references for the purpose or they may have insufficiently elaborated on the information present in the proposed references. If there is no dose because the study is observational, the range of concentrations for which an effect was noted relative to the comparator should be specified and entered.

Response: In “Table 1 and 3”, table columns were classified by each type of samples “Human, Animals, and In vitro”, it looked as if data was missing. Therefore, we aligned them into table rows and cells only have data from references has been described in the tables. By doing so, we could eliminate blank cells in the tables. In addition, cells from references only described “Chemical analysis” and “(no dose)” have also been deleted.

3. Some units of measurement suggest that the authors are unfamiliar with the quantity in question. μM/L is a meaningless unit of measurement. It should be μM or μmol/L.

Responses: All of units wrote as “μM/L” have been corrected to “μM” in all tables.

4. The topics of interest have been included in the tables but not enough outcomes and markers.

Responses: A column “Main findings and markers” has been added in “Table 1 and 3”. In “Table 3”, on the other hand, as the table was classified by the field “Diseases”, the field “Effects” shown in the “Table 1” has been deleted to simplify the table. In accordance with this change, some references and columns have been moved.

Changes with the addition of new references in tables and text:

Addition to the tables: (Please see in the manuscript)

Table 1 and 3.

Quercetin/Obesity prevention, liver function improvement (Table 1 only)/Visceral fat area (VFA) in low HLD subjects ↓ /9g/day: 12w/Human (RCT)/Nishimura M, 2019 [20]

Table 3.

Sepsis/Vitamin D/Correlation of vitamin D and LL-37 levels/Placebo, 200,000 IU, 400,000 IU: 24h within severe sepsis or septic shock/Human (RCT)/Quraishi SA, 2023 [153]

(We changed this reference because this reference was more suitable to the table.)

Moved within the table:

Table 3.

Dementia/Hesperidin/Cognitive function↑, Executing function↑, Episodic memory↑/32 or 275mg/day:  8w /Human/Hajialyani M, 2019 [25]

Dementia/Nobiletin/AD pathology↑, Motor function↑, Cognitive function↑, Aβ↓, Tau hyperphosphorylation↓/10-50mg/kg, i.p. or p.o./Mouse.rat/Nakajima A, 2019 [30]

Additional references:

20. Nishimura M.; Muro T.; Kobori M.; Nishihira J. Effect of Daily Ingestion of Quercetin-Rich Onion Powder for 12 Weeks on Visceral Fat: A Randomised, Double-Blind, Placebo-Controlled, Parallel-Group Study. Nutrients 2019, 12, 91.

140. Schafer MJ.; White TA.; Iijima K.; Haak AJ.; Ligresti G.; Atkinson EJ.; Oberg AL.; Birch J.; Salmonowicz H.; Zhu Y.; Mazula DL.; Brooks RW.; Fuhrmann-Stroissnigg H.; Pirtskhalava T.; Prakash YS.; Tchkonia T.; Robbins PD.; Aubry MC.; Passos JF.; Kirkland JL.; Tschumperlin DJ.; Kita H.; LeBrasseur NK. Cellular senescence mediates fibrotic pulmonary disease. Nat. Commun. 2017, 8, 14532.

144. Wautier JL.; Wautier MP. Pro- and Anti-Inflammatory Prostaglandins and Cytokines in Humans: A Mini Review. Int. J. Mol. Sci. 2023, 24, 9647.

153. Quraishi SA.; De Pascale G.; Needleman JS.; Nakazawa H.; Kaneki M.; Bajwa EK.; Camargo CA Jr.; Bhan I. Effect of Cholecalciferol Supplementation on Vitamin D Status and Cathelicidin Levels in Sepsis: A Randomized, Placebo-Controlled Trial. Crit. Care. Med. 2015, 43, 1928-1937.

Additional description to the text:

The new passage “(line-563) A randomised trial shows the patients with septic shock and sever sepsis rapidly induced LL-37 expression after the vitamin D supplementation [153].” has been added.

In addition, “(line-14) Abstract” have been modified. (Please see in the manuscript)

Other changes

“(line-95) 2. Brief methodology of the review article

     “Briefly, we explain the methodology of the search method of scientific papers in the references of this review. This review is narrative review as we described above. We searched for papers in PubMed with combined keywords “phytochemicals”, “vitamin D” and/or the name of each phytochemical such as “polyphenol”, “quercetin”, and the name of disorders like “lifestyle-related diseases”, “gut microbiota”, “lung disease” and/or “sepsis”. After the search, we composed of this review article in reference to papers we considered necessary. Furthermore, over 50% of references are published within 5 years other than we thought important or well-known papers in the field. References include our previous publications and we have searched and collected scientific papers that are attractive especially in our region, including papers introduced in the press releases of universities in Japan, webpages from companies and institutions, and papers in the web news of featured special articles in related field from newspapers. References in this review include many “Review” papers as they summaries many research findings.

     In section 3, we summarised the experimental information that was conducted in the referenced papers regarding to phytochemicals contained in fruits and vegetables into table 1. We searched on PubMed for the keywords “grape and polyphenol” with article types “Clinical trial and Randomised Controlled Trial” for two years from 2021 to 2022 in addition to 2023 before this research. As a result, 16 search results were emerged and they are summarised in the table 2. Then, table 3 shows references organised by diseases in section 4 and benefits of phytochemicals and vitamin D consumption. Since table 3 contains some references show in table 1, the item “effects” was deleted, and they were re-sorted by “diseases”. Since there are many review articles in the reference papers, references that do not include detailed experimental methods or doses are excluded from the table.”

With this regard. the 3rd paragraph in “(line-245) 3.7 Grapes” has been changed from ...

     “Table 2 summarises the clinical trials related with grape polyphenols. We searched on PubMed for the keywords “grape and polyphenol” with article types “Clinical trial and Randomised Controlled Trial” for two years from 2021 to 2022 in addition to 2023 before this research, and 16 search results were emerged [62 - 76]. In addition to cardiovascular disorders, recently conducted variety of randomised controlled trials for metabolic syndrome related disorders including dementia, gut microbiota related research, periodontal diseases, allergic diseases, and cancer revealed intriguing findings. [63 - 77].” 

to ...

     “(line-284) Table 2 shows the clinical trials related with grape polyphenols. In addition to cardiovascular disorders, recently conducted variety of randomised controlled trials for metabolic syndrome related disorders including dementia, gut microbiota related research, periodontal diseases, allergic diseases, and cancer revealed intriguing findings [63 - 77].“.

We sincerely hope our manuscript will be accepted and published in International Journal of Molecular Sciences, with your review.

We are looking forward to hearing from you at your earliest convenience.

Yours Sincerely,

Round 2

Reviewer 2 Report (Previous Reviewer 2)

As far as I am concerned, the manuscript has been adequately improved. I have no further comments.

This manuscript is a resubmission of an earlier submission. The following is a list of the peer review reports and author responses from that submission.

Round 1

Reviewer 1 Report

Thank you for re-submit the manuscript.This review discusses the potential of phytochemicals with vitamin D to prevent disease and promote healthy lifestyles.

This version is ok for me.

Author Response

Dear Sir/Madan,

Response: Thank you very much for your comment. We are happy to see your comments.

We are looking forward to seeing our manuscript to be published.

Yours Sincerely,

Reviewer 2 Report

The manuscript offers a revised and expanded version of a previously submitted paper. The review aimed to describe the beneficial effects of antioxidant molecules of plant origin and vitamin D.

I think that to deserve publication in the journal, the tables should be organized more scientifically and contain more detailed information: year of publication and first author, type of work (in vitro, animal model or human), study design (parallel, crossover, longitudinal), number of subjects involved, type of subjects involved (healthy, with pathologies, etc.), markers, outcomes, final results, etc.

Moreover, the Authors must explain why they are discussing Phytocvhemicals AND vitamin D? What link they are referring to? This connection seems arbitrary. What is the literature gap that the authors want to clarify? Prevention of chronic diseases? Infectious diseases? Interactions with the microbiota? Anti-inflammatory effect? Covid-19 pandemic? The parts of the manuscript seem to be unrelated to each other.

Author Response

Dear Sir/Madam,

Thank you very much for your comments and suggestions. We have made major changes in accordance with your comments.

Comments and Suggestions for Authors
The manuscript offers a revised and expanded version of a previously submitted paper. The review aimed to describe the beneficial effects of antioxidant molecules of plant origin and vitamin D.
I think that to deserve publication in the journal, the tables should be organized more scientifically and contain more detailed information: year of publication and first author, type of work (in vitro, animal model or human), study design (parallel, crossover, longitudinal), number of subjects involved, type of subjects involved (healthy, with pathologies, etc.), markers, outcomes, final results, etc.

Response: Thank you very much for your review comments. In accordance with your comment, we added “type of work, year of publication, and first author” in tables. Furthermore, we newly created “Table 2. Summary of randomised controlled trial using grape polyphenol (2021 to current)”. In accordance with the change, we added new paragraph in “2.7 Grapes”.

Changes:

We added new paragraph “(line-252) Table 2 summarises the clinical trials related with grape polyphenols. We searched on PubMed for the keywords !grape and polyphenol” with article types !Clinical trial and Randomised Controlled Trial” for two years from 2021 to 2022 in addition to 2023 before this research, and 16 search results were emerged [62 - 76]. In addition to cardiovascular disorders, recently conducted variety of randomised controlled trials for metabolic syndrome related disorders including dementia, gut microbiota related research, periodontal diseases, allergic diseases, and cancer revealed intriguing findings. These clinical trials have been confirmed the benefits of the consumption of polyphenol phytochemicals for their risk reducing effects in variety of disorders.”.

Moreover, the Authors must explain why they are discussing Phytochemicals AND vitamin D? What link they are referring to? This connection seems arbitrary. What is the literature gap that the authors want to clarify? Prevention of chronic diseases? Infectious diseases? Interactions with the microbiota? Anti-inflammatory effect? Covid-19 pandemic? The parts of the manuscript seem to be unrelated to each other.

Response: To summarise why we discuss phytochemicals and vitamin D in this review, we added passages in “Abstract” “(line-31) The intake of phytochemicals and vitamin D enhance anti- inflammatory effects, upregulate immunity, and reduce the risk of chronic disorders through maintaining healthy gut microbiota. Evidence during COVID-19 pandemic revealed that underlying conditions based on lifestyle-related diseases increase the risk of infectious diseases. Therefore, daily improvement and prevention of these conditions is very important. This narrative review discusses the importance of the intake of phytochemicals and vitamin D for the healthy lifestyle and preventing ME-BYO, non-disease conditions.” and showed these each part of manuscript has correlation.

In addition, we also added new passages in “4. Conclusions” “(line-561) The intake of phytochemicals and vitamin D enhance anti-inflammatory effects, upregulate immunity, and prevent chronic disorders through the maintenance of gut microbiota. Evidence during COVID-19 pandemic revealed that possessing underlying conditions such as lifestyle-related diseases induced by chronic inflammation increase the risk of infectious diseases. Therefore, daily improvement and prevention measure of these underlying conditions is very important. Our research predicts the necessity of the intake of phytochemicals and vitamin D for the healthy life, prevention of ME-BYO and variety of diseases in the near future.”.

Please see a marked-up copy with yellow highlighting of any changes in the new document in" !Revised Manuscript with Track Changes (supplementary)” file.

We sincerely hope the revisions we made will meet the requests, and the revised manuscript is now suitable for publication in International Journal of Molecular Sciences.

Yours Sincerely,

Round 2

Reviewer 2 Report

The Authors have improved the manuscript sufficiently

Author Response

RESPONSES TO REVIEWERS

Response to Reviewer 2 (ijms-2505842, Round 2):

Comments and Suggestions for Authors

The Authors have improved the manuscript sufficiently

Thank you very much for your comment. We received email from editorial office below.

>Significant revisions or new data are required before the manuscript can be considered for publication in IJMS.

>In particular the academic editor stated this "The manuscript has been improved according to incorporated changes and the evaluation of reviewers, but still, part of the information presented can be improved to have a higher impact on readers and applicability of the information in the review for further research. In the three tables, 1, 2, and 3, there is a key information missing: The dosages, The mg/day, or mg/kg of animal, or g/day in diet, etc., of each and every formulation/product/phytochemical and vitamin D form. This represents a big task to carry out in all the tables, but it would definitely improve the final result."

In accordance with the instruction from editorial office, we changed the field of “Type of work” to “Type of subjects and typical dose” and subfield “Human, Animals, and in vitro to “Table 1 and 3” were added. For “Table 2” we added the field “Dose”. We believe the new tables contain sufficient improvement for our manuscript. Please see them in our manuscript.

We sincerely hope our manuscript will be accepted and soon published in International Journal of Molecular Sciences.

Yours Sincerely,
